# Prevalence and Risk Factors of Pregnancy-Specific Urinary Incontinence: Findings from the Diamater Cohort Study

**DOI:** 10.3390/healthcare13172141

**Published:** 2025-08-28

**Authors:** Henrique Caetano Mingoranci Bassin, Angélica Mércia Pascon Barbosa, Caroline Baldini Prudencio, Luis Sobrevia, Vitoria Pascon Barbosa, Sthefanie Kenickel Nunes, Patrícia de Souza Rossignoli, Cristiane Rodrigues Pedroni, Danielle Cristina Honório França, Bruna Bologna Catinelli, Carolina Neiva Frota De Carvalho, Raissa Escandiusi Avramidis, Adriely Bittencourt Morgenstern Magyori, Carlos Izaias Sartorao-Filho, Marilza Vieira Cunha Rudge

**Affiliations:** 1Department of Gynecology and Obstetrics, Botucatu Medical School, São Paulo State University (Unesp), Botucatu 18618-000, Brazil; henrique-bassin@hotmail.com (H.C.M.B.); angelicapascon@gmail.com (A.M.P.B.); caroline_baldini@hotmail.com (C.B.P.); sobrevia@me.com (L.S.); sthe.kenickel@hotmail.com (S.K.N.); danielle.franca@unesp.br (D.C.H.F.); bologna.bruna@gmail.com (B.B.C.); carol_neiva@hotmail.com (C.N.F.D.C.); raissa_mids@hotmail.com (R.E.A.); adriely.morgenstern@gmail.com (A.B.M.M.); carlos.sartorao@unesp.br (C.I.S.-F.); 2Department of Physiotherapy and Occupational Therapy, School of Philosophy and Sciences, São Paulo State University (Unesp), Marilia 17525-900, Brazil; patricia.rossignoli@unesp.br (P.d.S.R.); cristiane.pedroni@unesp.br (C.R.P.); 3Cellular and Molecular Physiology Laboratory (CMPL), Department of Obstetrics, Division of Obstetrics and Gynaecology, School of Medicine, Faculty of Medicine, Pontificia Universidad Católica de Chile, Santiago 8330024, Chile; 4Medical School, Marília University (UNIMAR), Marília 17525-902, Brazil; vitoriapasconbarbosa@hotmail.com

**Keywords:** pregnancy complications, urinary incontinence, pregnancy, prevalence

## Abstract

**Introduction and Hypothesis:** Pregnancy-specific urinary incontinence (PS-UI) is defined as any onset of new urinary leakage during pregnancy. The study aims to analyze the prevalence and risk factors of PS-UI. We hypothesized that demographic and clinical factors, including Gestational Diabetes, may contribute to the development of PS-UI. **Methods:** We recruited pregnant women from the Diamater cohort study. We evaluated the maternal characteristics, demographics, anthropometrics, hyperglycemic status, and the PS-UI occurrence. A logistic regression model was performed considering the clinical variables to determine the predictive factors for PS-UI occurrence. **Results:** PS-UI was prevalent in 62.1% of women. Among these, 58.85% began leaking urine between 24–28 gestational weeks. Additionally, 51% developed PS-UI at the end of pregnancy. The pregestational BMI is a risk factor for PS-UI, and physical activity is a protective factor that halves the risk of PS-UI developing. **Conclusions:** Weight management and encouragement to engage in physical activity during pregnancy should be emphasized in prenatal care to reduce the risk of PS-UI. Further studies are suggested to evaluate the impact of the association due to UI later in life.

## 1. Introduction

Urinary incontinence (UI) is a prevalent condition among women worldwide (25–45%) that can cause physical and emotional distress, have financial consequences, and place a burden on individuals and society [1,2]. Women with UI consume significantly more medical resources and incur higher costs than those without UI, but standardized prevention and treatment may positively impact costs and outcomes [3].

Pregnancy-specific urinary incontinence (PS-UI) is UI that first appears during pregnancy [4]. Although PS-UI affects up to half of pregnant women and peaks in the third trimester [5,6,7,8,9], there are limited studies on its epidemiology. PS-UI is a strong predictor of UI postpartum and later in life, and identifying its risk factors will help healthcare providers and pregnant women make informed decisions [2,10,11].

PS-UI may be mild to moderate, but UI can negatively impact health-related quality of life [8,12]. However, few pregnant women seek professional assistance for urinary leakage [12,13]. Prenatal care is an entry point into the health system and provides an opportunity to engage women and provide information about disease severity [14]. Few integrated care models link antenatal care and non-communicable disease prevention [14,15] despite their effectiveness in communicable disease prevention [16]. Opportunistic screening for PS-UI risk factors may be feasible during critical points in the life course [14]. Previous studies have identified obesity, parity, and vaginal delivery as key risk factors for PS-UI [2,11]. However, the role of gestational diabetes and physical activity remains less understood.

Our study aims to analyze the prevalence and risk factors of PS-UI during the pregnancy span, taking into account life-course approaches to women’s health. We hypothesize that demographic and clinical factors, such as obesity, sedentarism, smoking, and gestational diabetes mellitus (GDM), may contribute to the development of PS-UI.

## 2. Method

### 2.1. Research Design and Subjects

The DIAMATER study is an ongoing prospective cohort study that examines the link between GDM and pelvic floor muscle dysfunction as moderators between GDM and UI throughout the lifespan [17]. Following the Declaration of Helsinki, we conducted and approved the study by the Institutional Ethical Committee of the Botucatu Medical School of São Paulo State University (Protocol Number CAAE 82225617.0.0000.5411). Participants provided written informed consent, and their data were kept confidential. This report was consistent with the STROBE statement. This observational study screened 1450 pregnant women at the Perinatal Diabetes Research Center (PDRC) at the University Clinical Hospital of Botucatu Medical School (UNESP), Brazil, and followed them until delivery. All patients received prenatal care and gave birth at the same institution. Women were selected from the DIAMATER cohort. Recruitment was conducted at the Perinatal Diabetes Research Center through routine prenatal visits. Eligible participants met the inclusion criteria described below and consented to participate, ensuring a consecutive, non-randomized sample representative of the clinic’s patient population.

Inclusion criteria: adult (age 18–40) women in their first pregnancy and women in the second pregnancy, who had a prior planned C-section in their previous pregnancy, to avoid the impact of previous parturition and vaginal birth on the pelvic floor function. While prior vaginal delivery is a recognized risk factor for UI, we included primiparous women only if they had a prior planned C-section to minimize pelvic floor trauma. However, we acknowledge that pregnancy itself may alter pelvic floor mechanics, and thus stratified analyses by parity. All participants in the study began receiving prenatal care at the first trimester of gestation and underwent a hyperglycemia screening test at the first trimester and between 24–28 weeks of pregnancy, according to the actual recommendation screening and diagnosis [17,18,19].

Women with pregestational UI, known type 1 or type 2 diabetes, preterm delivery (<37 weeks of gestation), multiple pregnancies, known fetal anomaly, connective tissue diseases, and any clinical condition that may have jeopardized their health status were excluded from the study. Additionally, women with oncological, neurodegenerative, or autoimmune diseases were excluded due to their potential confounding effects on pelvic floor function. Pregestational diabetes (types 1/2) was excluded because it represents a distinct metabolic state with potential long-term effects on pelvic floor integrity, whereas gestational diabetes reflects pregnancy-specific metabolic changes that may differently influence UI risk.

### 2.2. Data Collection

Participants were recruited until 24 weeks of gestation and invited to join the study if they met the inclusion criteria. We recruited 992 participants between 2017 and 2022. After giving their written consent, they were asked to answer a questionnaire with personal details; clinical and obstetric, historical, and anthropometric measures were taken. Participants were evaluated at two time points (TP): 24–28 weeks of gestation (1st TP) and 36–38 weeks of gestation (2nd TP).

PS-UI was defined as any onset of new urinary leakage during pregnancy [4]. The participants were asked to answer “yes” or “no” as to whether they had experienced PS-UI. Participants who gave positive responses were identified as having UI, following the definition set by the International Continence Society [20]. Early PS-UI was considered to be incontinence that started before 28 weeks of gestation, and Late PS-UI was that which started at the end of the third trimester, after 36 weeks of gestation. PS-UI was defined as any new onset of urinary leakage (≥1 episode) during pregnancy, confirmed via a single ‘yes/no’ question. While this captures prevalence, it does not differentiate severity, frequency, or impact on quality of life. The questionnaire did not differentiate between stress urinary incontinence (SUI), urge urinary incontinence (UUI), or mixed urinary incontinence (MUI). Therefore, our prevalence estimates refer to total UI as defined by the International Continence Society, without subtyping.

Physical activity was self-reported as any structured exercise performed regularly during pregnancy. However, frequency, duration, and intensity were not assessed.

In this cohort, the diagnostic guidelines proposed by the American Diabetes Association were used to identify patients with GDM [19] using the 75 g oral glucose tolerance test (75 g-OGTT).

### 2.3. Statistical Analysis

The prevalence of PS-UI was calculated by obtaining the frequencies of UI in the 1st TP, 2nd TP (without intersections), and 1st TP plus 2nd TP. With these three groups, interactions with the demographic variables were made using the chi-square test.

A logistic regression model was performed considering the occurrence of UI in these three scenarios, including demographic variables as explanatory variables, to determine the participants’ risk or protective factors for UI. We used a significant 5% or the corresponding p-value in all tests. All analyses were performed using SAS for Windows, version 9.4 (SAS Institute Inc., Cary, NC, USA), including descriptive statistics, chi-square tests, and logistic regression modeling.

## 3. Results

Among 992 pregnant women in the Diamater cohort (Figure 1), 62.1% (*n* = 616) developed PS-UI while 37.9% (*n* = 376) did not. Table 1 shows the demographic characteristics by PS-UI status.

Compared to those without PS-UI, women with PS-UI had a higher pregestational BMI (*p* = 0.002) and BMI at the 1st TP (*p* = 0.004) and 2nd TP (*p* = 0.002). Moreover, women with PS-UI engaged in less physical activity during pregnancy (*p* = 0.003) and presented with more chronic coughing (*p* = 0.025).

Although pregestational BMI differed significantly between groups, gestational weight gain at both the first and second time points did not differ significantly. This suggests that baseline BMI, rather than weight gain during pregnancy, was the primary anthropometric risk factor for PS-UI in our cohort.

PS-UI prevalence among the studied population was 62.1%, with 58.85% occurring between 24 and 28 weeks of gestation, as shown in Table 2. Excluding the 83 cases of early PS-UI (Figure 1), the prevalence of late PS-UI among pregnant women was 51%. Percentages for early (58.85%) and late (51%) PS-UI are calculated independently for each time point and are not mutually exclusive. Some women who experienced early PS-UI also reported symptoms in late pregnancy; therefore, the sum exceeds 100%.

PS-UI, Pregnancy-Specific Urinary Incontinence; BMI, body mass index; OGTT, oral glucose tolerance test; TP, time point (first: 24–28 weeks of gestation; second: 36–38 weeks of gestation). Data are expressed as means ± standard deviations or absolute frequency (n) and percentage (%). The differences between the groups were compared using the Chi-square test. Significance *p* < 0.05. Percentages are based on the number of patients responding to each question.

Both early and late incontinence were associated with higher pregestational BMI and BMI from the time point it was assessed, as shown in Table 3. Women with early PS-UI engaged in less physical activity during pregnancy (*p* = 0.004), while those with late PS-UI reported more chronic coughing (*p* = 0.012) and alcohol consumption during pregnancy (*p* = 0.047).

As demonstrated in Table 4, the logistic regression model identified pregestational BMI as a risk factor for PS-UI (OR = 1.04, IC95%: 1.01–1.07). Physical activity was identified as a protective factor, halving the risk of developing PS-UI (OR = 0.5, IC95%: 0.32–0.80). The analysis of the 1st TP also revealed that pregestational BMI was a risk factor and physical activity was a protective factor for Early PS-UI. There was no correlation between the variables and the incidence of Late PS-UI.

## 4. Discussion

PS-UI was prevalent in 62.1% of pregnant women, 58.85% began leaking urine between 24 and 28 gestational weeks, and 51% developed PS-UI at the end of pregnancy. The pregestational BMI is a risk factor for PS-UI, and physical activity is a protective factor that halves the risk of PS-UI developing.

A systematic review by Pizzoferrato et al. (2023) [21] showed that primiparous women had a higher risk of postpartum UI following vaginal delivery compared to C-section, regardless of their continence status before childbirth. Nonetheless, the protective effect of C-section was significantly higher in nulliparous women without UI before delivery. Compared to our results, we detected a higher prevalence of PS-UI even in a sample with a previous C-section [21].

A literature review of lower urinary tract symptoms in women, with a particular emphasis on incontinence and overactive bladder, revealed that age, smoking, pregnancy, asthma, obesity, dementia, vaginal delivery, constipation, diuretic use, and certain medications were identified as risk factors for UI [22].

The pathophysiology underlying antenatal and postpartum UI remains poorly understood [23]. The reduced protective effect of C-section against antenatal UI in nulliparous women with UI may be attributed to the characteristics of the women’s tissues [24]. It is possible to assume that preexisting UI or pregnancy-induced UI may not be associated with perineal trauma but rather with weakened supportive tissues. Thus, it could clarify why a C-section may not prevent UI in such cases [21].

A recent study that evaluated pelvic floor outcomes at different stages of pregnancy, which included stress incontinence, anal incontinence, prolapse, and sexual dysfunction, found that at least one pelvic floor disorder symptom was experienced by 60.8% of the study cohort during pregnancy and that particular symptoms were exacerbated in the third trimester of pregnancy. The prevalence of PS-UI was lower than that presented in this study, with the participants’ overall rates equally distributed in the first, second, and third trimesters of pregnancy (*p* = 0.168). In the third trimester of pregnancy, symptoms related to urinary distress were reported to be more intense than those in the first and second trimesters of pregnancy [25]. The greater intensity of urinary distress in the third trimester may be explained by the cumulative effects of increasing fetal weight, higher intra-abdominal pressure, and progressive stretching/weakening of pelvic floor structures.

Obesity is a well-established risk factor for UI, regardless of sex, age, and other factors [22]. Hence, obesity may lead to UI in several ways. Firstly, excess weight puts additional pressure on the pelvis, bladder, and pelvic floor muscles, weakening these muscles and impairing the ability to control urination. Additionally, obesity may also increase systemic inflammation and intra-abdominal pressure, which can lead to bladder dysfunction and UI [26]. Chronic coughing, a significant factor in late PS-UI (*p* = 0.012), may exacerbate intra-abdominal pressure, straining the pelvic floor. We hypothesize that repeated coughing-induced stress could weaken pelvic floor muscles, similar to mechanisms seen in obesity-related UI [26]. Future studies should measure cough frequency/intensity and its biomechanical effects.

That weight loss may help improve UI in people with obesity. A study conducted by Subak et al. (2005) found that a structured weight loss program led to a significant reduction in UI episodes in overweight and obese women [27]. Similarly, a meta-analysis reported that weight loss interventions were effective in reducing UI in women with obesity [28].

The exclusion of women with pregestational UI or prior vaginal deliveries limits the generalizability of our findings to nulliparous women and those with planned C-sections. Future studies should include multiparous women and diverse delivery histories to assess PS-UI risk across broader populations.

The binary PS-UI definition may overestimate clinically significant incontinence. Recurrent episodes or severity metrics would improve specificity.

Future longitudinal studies should track PS-UI progression postpartum, correlating pregnancy-specific risk factors (e.g., BMI, physical activity) with long-term urinary and pelvic floor health. Standardized tools like ICIQ-UI SF could quantify severity and subtype-specific trajectories.

Stratified analyses suggested parity did not significantly modify PS-UI risk factors, though larger samples are needed to confirm this.

## 5. Conclusions

Most women had urinary incontinence at some point during their pregnancy. The onset of PS-UI was proportional among those who leaked urine between 24 and 28 gestational weeks and those who leaked at the end of pregnancy. The pregestational BMI is a risk factor for PS-UI, and physical activity is a protective factor that halves the risk of PS-UI developing. Targeted physical activity programs, such as aerobic exercises and pelvic floor training, should be investigated for their efficacy in reducing PS-UI risk. Clinical guidelines could integrate such interventions, tailored to pregestational BMI and trimester-specific needs. While weight control and physical activity during pregnancy are important to reduce PS-UI and potentially lower the risk of UI later in life, preventive strategies should begin earlier—in adolescence, young adulthood, and pre-conception care—to optimize pelvic floor health across the lifespan. While this study identifies risk factors for PS-UI during pregnancy, it does not assess long-term outcomes. Further prospective studies are needed to evaluate the persistence of PS-UI postpartum and its impact on quality of life, pelvic floor dysfunction, and healthcare utilization in later years.

## Figures and Tables

**Figure 1 healthcare-13-02141-f001:**
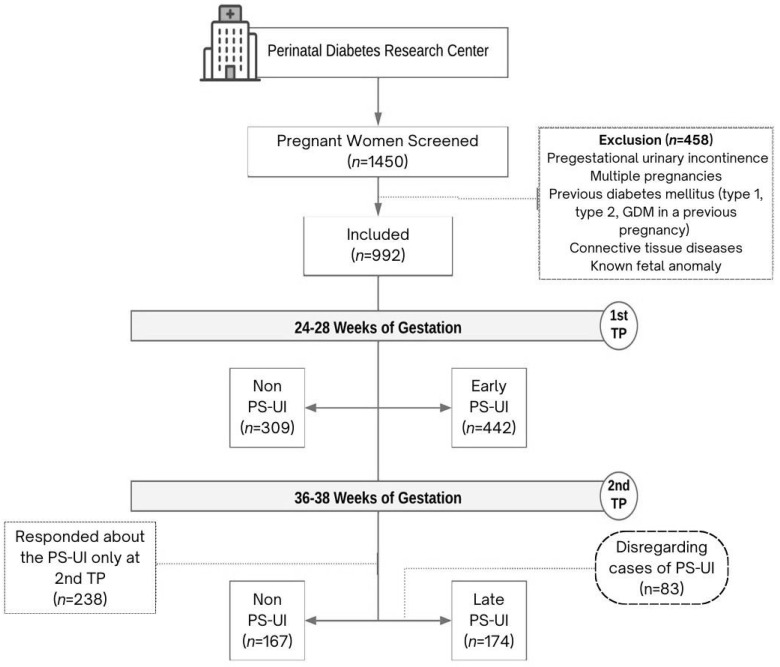
Participant flow diagram showing screening, enrollment, and PS-UI classification in the Diamater cohort study.

**Table 1 healthcare-13-02141-t001:** Characteristics of the study population.

Variable			Non-PS-UI	PS-UI	*p*-Value
Partnership status		Married	311 (83.4%)	515 (83.6%)	0.926
		Not married	62 (16.6%)	101 (16.4%)	
Education level		basic level	27 (7.2%)	42 (6.9%)	0.014
		high school	229 (61.1%)	427 (69.7%)	
		college/university	119 (31.7%)	144 (23.5%)	
Ethnicity		Caucasian	299 (81%)	495 (80.5%)	0.835
		Non-Caucasian	70 (19%)	120 (19.5%)	
Pregestational BMI (kg/m^2^)			26.9 ± 6.25	28.48 ± 7.28	0.002
BMI–1st TP (kg/m^2^)			29.4 ± 6.04	30.8 ± 6.85	0.004
BMI–2nd TP (kg/m^2^)			31.6 ± 6.20	33.75 ± 6.51	0.002
Gestational weight gain–1st TP (kg)			6.39 ± 5.04	6.08 ± 6.19	0.477
Gestational weight gain–2nd TP (kg)			11.76 ± 7.34	11.36 ± 7.63	0.613
Fasting blood glucose (mg/dL)			79.48 ± 12.71	80.11 ± 14.44	0.583
OGTT–fasting (mg/dL)			74.82 ± 12.66	76.11 ± 11.06	0.161
OGTT–1 h (mg/dL)			116.7 ± 31.27	119.17 ± 32.7	0.321
OGTT–2 h (mg/dL)			105.08 ± 28.1	107.01 ± 8.88	0.378
Chronic coughing			0	9 (1.8%)	0.025
Constipation			91 (31.5%)	152 (28.9%)	0.439
Fecal incontinence			3 (1.1%)	6 (1.2%)	0.887
Previous arterial hypertension			24 (8.3%)	53 (10.2%)	0.376
Alcohol consumption			1 (0.4%)	9 (1.8%)	0.091
Smoking in pregnancy			12 (4.3%)	26 (5%)	0.615
Physical activity			66 (23.1%)	76 (14.7%)	0.003
Gestational diabetes mellitus			81 (22%)	154 (25.2%)	0.242
Pregnancy-induced hypertension			2 (3%)	8 (4.4%)	0.611
Urinary tract infection			7 (10.8%)	22 (12.4%)	0.735

PS-UI, Pregnancy-Specific Urinary Incontinence; BMI, body mass index; OGTT, oral glucose tolerance test; TP, time point (1st: 24–28 weeks of gestation; 2nd: 36–38 weeks of gestation); Data are presented as mean ± SD or n (%). Percentages reflect response rates for each variable. The differences between the groups were compared using the Chi-square test. Significance *p* < 0.05. Percentages are based on the number of patients responding to each question. As some participants did not answer every question, percentages were calculated using the number of valid responses per item rather than the total cohort size.

**Table 2 healthcare-13-02141-t002:** Prevalence of Pregnancy-Specific Urinary Incontinence.

Weeks of Gestation			
		*n*	%
**24–28 weeks**			
1st TP	(1) non-PS-UI	309	41.15
	(2) Early PS-UI	442	58.85
**36–38 weeks**			
2nd TP	(1) non-PS-UI	167	49.0
	(2) Late PS-UI	174	51.0
**1st and 2nd TP**			
	(1) non-PS-UI	376	37.9
	(2) PS-UI	616	62.1

PS-UI, Pregnancy-Specific Urinary Incontinence; TP, time point (1st: 24–28 weeks of gestation; 2nd: 36–38 weeks of gestation); Data are expressed as absolute frequency (*n*) and percentage (%).

**Table 3 healthcare-13-02141-t003:** Association between Pregnancy-Specific Urinary Incontinence (PS-UI) and characteristics according to the time points.

Variable		1st TP			2nd TP		
		Non-PS-UI (*n* = 309)	Early PS-UI (*n* = 442)	***p*-Value**	Non-PS-UI (*n* = 167)	**Late PS-UI (*n* = 174)**	***p*-Value**
Partnership status	Married	261 (84.5%)	367 (83%)	0.601	136 (82.4%)	148 (85.1%)	0.511
	Not married	48 (15.5%)	75 (17%)		29 (17.6%)	26 (14.9%)	
Education level	basic level	22 (7.1%)	27 (6.1%)	0.007	11 (6.6%)	15 (8.7%)	0.210
	high school	191 (61.8%)	319 (72.5%)		93 (55.7%)	108 (62.4%)	
	college/university	96 (31.1%)	94 (21.4%)		63 (5.2%)	50 (28.9%)	
Ethnicity	Caucasian	253 (82.4%)	365 (82.6%)	0.952	124 (76.1%)	130 (75.1%)	0.843
	Non-Caucasian	54 (17.6%)	77 (17.4%)		39 (23.9%)	43 (24.9%)	
Pregestational BMI (kg/m^2^)		27.04 ± 6.28	28.5 ± 7.33	0.003	27.31 ± 6.42	29.47 ± 6.69	0.003
BMI–1st TP (kg/m^2^)		29.49 ± 6.1	30.83 ± 6.87	0.005	29.58 ± 6.8	30.24 ± 6.64	0.637
BMI–2nd TP (kg/m^2^)		31.23 ± 6.76	33.49 ± 6.33	0.213	31.67 ± 6.2	33.92 ± 6.64	0.002
Gestational weight gain–1st TP (kg)		6.31 ± 5.01	6.09 ± 6.26	0.614	6.18 ± 7.73	5.85 ± 4.97	0.833
Gestational weight gain–2nd TP (kg)		10.81 ± 5.55	10.62 ± 8.76	0.863	11.6 ± 8.18	11.81 ± 6.83	0.802
Fasting blood glucose (mg/dL)		79.86 ± 12.4	79.42 ± 12.23	0.691	81.63 ± 13.26	82.12 ± 19.37	0.826
OGTT–fasting (mg/dL)		74.58 ± 12.83	75.77 ± 10.58	0.206	78.46 ± 11.73	77.84 ± 13.2	0.755
OGTT–1 h (mg/dL)		115.65 ± 29.98	118.24 ± 32.9	0.311	126.02 ± 35.13	123.81 ± 31.5	0.755
OGTT–2 h (mg/dL)		105.08 ± 26.77	106.19 ± 28.86	0.615	112.05 ± 33.39	111.15 ± 28.87	0.862
Chronic coughing		0	1 (0.3%)	0.426	0	8 (6.7%)	0.012
Constipation		79 (31.6%)	113 (28.7%)	0.430	27 (27.8%)	39 (29.5%)	0.778
Fecal incontinence		1 (0.4%)	3 (0.8%)	0.568	2 (2.2%)	3 (2.5%)	0.879
Previous arterial hypertension		17 (6.8%)	29 (7.3%)	0.795	12 (12.4%)	24 (19.4%)	0.163
Alcohol		1 (0.4%)	4 (1%)	0.391	0	5 (4.2%)	0.047
Smoking		8 (3.2%)	19 (4.8%)	0.328	6 (6.6%)	7 (5.9%)	0.832
Physical activity		53 (21.4%)	51 (12.8%)	0.004	22 (22.9%)	25 (20.7%)	0.689
Gestational diabetes mellitus		70 (23.0%)	103 (23.5%)	0.877	116 (70.7%)	121 (70.3%)	0.939
Pregnancy-induced hypertension		0	3 (4.2%)	0.185	3 (3.8%)	5 (4.6%)	0.778
Urinary infection		1 (2.6%)	7 (9.5%)	0.184	9 (11.3%)	15 (14.4%)	0.526

PS-UI, Pregnancy-Specific Urinary Incontinence; BMI, body mass index; OGTT, oral glucose tolerance test; TP, time point (1st: 24–28 weeks of gestation; 2nd: 36–38 weeks of gestation); Data are expressed as means ± standard deviations or absolute frequency (*n*) and percentage (%). The differences between the groups were compared using the Chi-square test. Significance *p* < 0.05. *p*-values represent the results from the relevant statistical tests. Percentages are based on the number of patients responding to each question.

**Table 4 healthcare-13-02141-t004:** Model of logistic regression model of Pregnancy-Specific Urinary Incontinence (PS-UI).

Variable		Early PS-UI (*n* = 442)				Late PS-UI (*n* = 442)				**PS-UI (1st plus 2nd TP)**			
		OR *	95% CI **		*p*-Value	OR *	95% CI **		*p*-Value	**OR ***	**95% CI ****		***p*-Value**
Education level	basic level												
	high school	1.28	0.62–2.63		0.172	0.62	0.04	9.28	0.491	1.21	0.58–2.51		0.304
	college/university	0.89	0.40–1–96		0.359	1.20	0.08–19.02		0.629	0.91	0.41–2.04		0.493
Ethnicity	Caucasian	1.22	0.76–1.96		0.416	2.65	0.46–15.15		0.274	1.28	0.88–0.78		0.519
	Non-Caucasian												
Age (years)		0.99	0.96–1.02		0.375	1.01	0.92–1.12		0.805	0.99	0.96–1.03		0.706
Pregestational BMI		1.03	1.01–1.06		0.014	1.02	0.93–1.11		0.716	1.04	1.01–1.07		0.006
Constipation		1.21	0.84–0.83		0.312	1.02	0.26–4.03		0.974	0.78	0.54–1.13		0.193
Fecal incontinence		3.02	0.29–31.82		0.358					2.81	0.26–29.92		0.392
Previous arterial hypertension		0.83	0.41–1.70		0.616	1.4	0.16–12.32		0.761	0.84	0.41–1.76		0.652
Smoking		2.08	0.78–5.53		0.141					1.85	0.69–4.91		0.220
Physical activity		0.51	0.32–0.80		0.003	0.72	0.13–3.88		0.700	0.5	0.32–0.79		0.003
Gestational diabetes mellitus		1.10	1.01–1.06		0.649	0.94	0.25–3.58		0.928	1.16	0.75–1.81		0.500

PS-UI, Pregnancy-Specific Urinary Incontinence; BMI, body mass index; TP, time point (1st: 24–28 weeks of gestation; 2nd: 36–38 weeks of gestation). * Odds Ratio; ** 95% Confidence Interval for Odds Ratio. Significance *p* < 0.05. *p*-values represent the results from the relevant statistical tests.

## Data Availability

The original contributions presented in this study are included in the article. Further inquiries can be directed to the corresponding author(s).

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
