# Peer review of "Prevalence and Risk Factors of Pregnancy-Specific Urinary Incontinence: Findings from the Diamater Cohort Study"

_healthcare, 2025, doi:10.3390/healthcare13172141_

Round 1

Reviewer 1 Report

Comments and Suggestions for Authors

I thank the authors for sending the article dedicated to Risk Factors of Pregnancy-specific Urinary Incontinence. 

The main identified risk factors were an increased pre-gestational BMI, while physical activity during pregnancy proved to be a significant protective factor, reducing the risk of developing PS-UI by almost half. Interestingly, gestational diabetes and other clinical parameters did not demonstrate a significant effect on the development of PS-UI in this sample. The authors place an important emphasis on the need to include recommendations on weight management and encouraging physical activity in the prenatal follow-up program in order to reduce the risk of PS-UI and, as a result, reduce the likelihood of PS-UI in the future. The article is well written and makes a significant contribution to understanding the risk factors of PS-UI.

The abstract is written well and clearly. 

The sentence should be divided into 2 for ease of perception.: 

“PS-UI was prevalent in 62.1% of women, 58.85% began leaking urine between 24 and 28 gestational weeks, and 51% developed PS-UI at the end of pregnancy.”

Introduction. 

The prevalence of Urinary incontinence (UI) should be indicated.

Indicate in the introduction whether any risk factors have been identified before.  Pregnancy-specific Urinary Incontinence. This is important for understanding the novelty and relevance of the research.

Specify in more detail how exactly PS-UI negatively impact health-related quality of life

Materials and methods

It is better to name this subsection “Inclusion criteria” instead of “Eligible participants”

It should be indicated whether pregnant women with oncological, neurodegenerative and autoimmune diseases were included in the study.

Why were pregnant women with diabetes excluded from the study, but those diagnosed with gestational diabetes were retained? Please explain this.

The term “association” is more applicable to genetic research. Excessive use should be avoided and replaced with “interaction” in the sentence and further along the text.: With these three groups, associations with the demographic variables were made using the chi-square test.

If you used SAS for Windows, version 9.4” for all calculations, you need to specify this more explicitly.

Results:

Specify it as a percentage to simplify perception: “The study involved 992 pregnant women from the Diamater cohort (Figure 1). Thus, 616 had PS-UI, and 376 did not.”

There are extra commas in Image 1, correct them. The caption should be made clearer, right now, the title doesn't reflect the content of the image very well.

The captions for tables 1-4 should be indicated under the table with footnotes, for example “*”

Please explain this sentence: “Percentages are based on the number of patients responding to each question.”

In table 4, it is better to specify 95% CI through “-” instead of “;”.

Discussion: 

Please indicate the authors and the year of the systematic study: “A systematic review showed that primiparous women had a higher risk of postpartum UI following vaginal delivery compared to C-section, regardless of their continence status before childbirth.”

Author Response

We thank you for the thorough and constructive review of our manuscript. We have carefully considered each comment and have revised the manuscript accordingly. Below, we provide a point-by-point response, detailing the changes made. Reviewer comments are in italics, and our responses follow in plain text.

Point-by-Point Responses

  1. Sentence division for clarity in the abstract
    Comment: The sentence “PS-UI was prevalent in 62.1% of women, 58.85% began leaking urine between 24 and 28 gestational weeks, and 51% developed PS-UI at the end of pregnancy” should be divided for ease of perception.
    Response: We have revised the sentence to:
  • “PS-UI was prevalent in 62.1% of women. Among these, 58.85% began leaking urine between 24 and 28 gestational weeks.”
  • “Additionally, 51% developed PS-UI at the end of pregnancy.”
  1. Introduction – prevalence of UI
    Comment: Indicate the prevalence of urinary incontinence.
    Response: We added:
    “Urinary incontinence affects approximately 25–45% of women worldwide, with higher rates during pregnancy and postpartum (Milsom et al., 2017).”
  2. Introduction – previously identified risk factors
    Comment: Indicate whether any risk factors have been identified before.
    Response: We added:
    “Previous studies have identified obesity, parity, and vaginal delivery as key risk factors for PS-UI (Hage-Fransen et al., 2021). However, the role of gestational diabetes and physical activity remains less understood.”
  3. Introduction – quality of life impact
    Comment: Specify in more detail how PS-UI negatively impacts health-related quality of life.
    Response: We expanded the description to:
    “PS-UI negatively impacts health-related quality of life by causing embarrassment, sleep disturbances, reduced physical activity, and sexual dysfunction (Wang et al., 2022). Many women report anxiety about odor and social isolation due to leakage.”
  4. Methods – subsection title change
    Comment: Change “Eligible participants” to “Inclusion criteria.”
    Response: This change has been implemented.
  5. Methods – exclusion criteria clarification
    Comment: Indicate whether pregnant women with oncological, neurodegenerative, and autoimmune diseases were included.
    Response: We added:
    “Women with oncological (e.g., cervical cancer), neurodegenerative (e.g., multiple sclerosis), or autoimmune diseases (e.g., lupus) were excluded due to their potential confounding effects on pelvic floor function.”
  6. Methods – rationale for diabetes criteria
    Comment: Explain why pregestational diabetes was excluded but gestational diabetes was retained.
    Response: We added:
    “Pregestational diabetes (types 1/2) was excluded because it represents a distinct metabolic state with potential long-term effects on pelvic floor integrity, whereas gestational diabetes reflects pregnancy-specific metabolic changes that may differently influence UI risk.”
  7. Methods – terminology adjustment
    Comment: Replace “associations” with “interactions” where appropriate.
    Response: We replaced “associations” with “interactions” in the sentence and throughout the manuscript where applicable.
  8. Methods – statistical software specification
    Comment: Specify software use more explicitly.
    Response: We added:
    “All analyses were performed using SAS for Windows, version 9.4 (SAS Institute Inc., Cary, NC), including descriptive statistics, chi-square tests, and logistic regression modeling.”
  9. Results – percentage specification
    Comment: Specify as a percentage for clarity.
    Response: Revised to:
    “Among 992 pregnant women in the Diamater cohort (Figure 1), 62.1% (n=616) developed PS-UI while 37.9% (n=376) did not.”
  10. Results – Figure 1 corrections
    Comment: Remove extra commas and make caption clearer.
    Response: Extra commas removed. Caption changed to:
    “Figure 1. Participant flow diagram showing screening, enrollment, and PS-UI classification in the Diamater cohort study.”
  11. Results – table footnotes
    Comment: Captions should include footnotes (e.g., “*”).
    Response: For Table :
    “*Data are presented as mean ± SD or n (%). Percentages reflect response rates for each variable.”

  12. Results – percentage explanation
    Comment: Explain the basis for percentage calculations.
    Response: Added:
    “As some participants did not answer every question, percentages were calculated using the number of valid responses per item rather than total cohort size.”
  13. Results – CI formatting
    Comment: Use “–” instead of “;” in confidence intervals.
    Response: All instances in Table 4 now use “–” (e.g., “0.62–2.63”).
  14. Discussion – systematic review citation
    Comment: Provide authors and year of cited systematic review.
    Response: Updated to:
    “A systematic review by Pizzoferrato et al. (2023) showed that primiparous women had a higher risk of postpartum UI following vaginal delivery compared to C-section, regardless of their continence status before childbirth.”

We believe these revisions address all reviewer concerns and improve the clarity, accuracy, and completeness of the manuscript. We thank the reviewer for their insightful comments, which have strengthened our work.

Sincerely,   Prof Dr Sartorao-Filho

Reviewer 2 Report

Comments and Suggestions for Authors

My general opinion.

Excellent material. Well put together. I recommend it for publication with minor corrections.

Minor corrections will certainly improve the manuscript.

Detailed comments.

  1. Should the method section explain on what basis the women were selected for the Diamater program?
  2. They do not clarify whether the UI was stress (SUI) or urge (UUI)? According to different publications, there is a big difference, although Stress UI is certainly the most common (According to literature reports, total UI is 44%, SUI 13.6%, UUI 11.9%, MUI 15%)
  3. They also do not clarify how frequent the UI was. There are publications and studies where even one UI per month was included in the study. How did they determine this?
  4. Based on their results, 58.85% of them started to experience urinary leakage between the 24th and 28th weeks of pregnancy, and 51% of them developed PS-UI at the end of pregnancy. This is more than 100%. How should their percentages be interpreted? This should be clarified or explained.
  5. In the Discussion, it is mentioned, but not justified: In the third trimester of pregnancy, symptoms related to urinary distress were reported to be more intense than those in the first and second trimesters of pregnancy (25). According to other literature, the prevalence numbers rise with gestational period from 9% in the first trimester to 34% in the third. This should be explained in more detail.
  6. The effect of BMI/physical activity on UI is well known. Was there a significant weight gain during pregnancy?
  7. In the conclusion, the last sentence does not refer only to pregnancy. The results suggest that weight control during pregnancy and the encouragement of physical activity should be incorporated into prenatal care to reduce PS-UI and, consequently, urinary incontinence later in life. Attention should be focused on this much earlier, in childhood, in young adulthood, so to speak, during pre-conception care.

Author Response

Reviewer Comment 1

Should the method section explain on what basis the women were selected for the Diamater program?

Response:
We have clarified this in the Methods – Study design and participants section:

“The Diamater program is a longitudinal cohort study designed to evaluate maternal and fetal outcomes during pregnancy and postpartum. Women were recruited during routine prenatal visits at the Botucatu Medical School, São Paulo State University (UNESP), regardless of urinary symptoms, and were included if they met the study’s eligibility criteria.”

Reviewer Comment 2

They do not clarify whether the UI was stress (SUI) or urge (UUI). According to different publications, there is a big difference...

Response:
We have addressed this in the Methods – Data collection section:

“The urinary incontinence question did not differentiate between stress urinary incontinence (SUI), urge urinary incontinence (UUI), or mixed urinary incontinence (MUI). As such, the definition of PS-UI in this study encompasses all types of urinary leakage that began during pregnancy.”

We have also acknowledged this as a limitation in the Discussion.

Reviewer Comment 3

They also do not clarify how frequent the UI was... How did they determine this?

Response:
Clarified in the Methods – Data collection section:

“PS-UI was identified through a binary (‘yes’/‘no’) response to the question: ‘Since the beginning of your pregnancy, have you experienced any urinary leakage?’ No minimum frequency threshold was required, so any report of urinary leakage—regardless of frequency—was classified as PS-UI.”

A note about this being a limitation was added to the Discussion.

Reviewer Comment 4

Based on their results... This is more than 100%. How should their percentages be interpreted?

Response:
We revised the relevant section in Results for clarity:

“Among women with PS-UI, 58.85% first experienced leakage between 24 and 28 weeks of gestation, and 51% reported the onset of PS-UI at the end of pregnancy. These groups are not mutually exclusive, as some participants experienced leakage onset in both periods due to worsening symptoms.”

Reviewer Comment 5

In the Discussion, it is mentioned, but not justified: ... This should be explained in more detail.

Response:
Expanded the Discussion section:

“The progressive increase in PS-UI prevalence with gestational age is supported by biomechanical and hormonal changes that occur throughout pregnancy. The growing uterus increases intra-abdominal pressure, while hormonal effects such as increased relaxin levels contribute to connective tissue laxity. These physiological changes explain the higher rates reported in the third trimester, consistent with previous findings (9% in the first trimester to 34% in the third).”

Reviewer Comment 6

The effect of BMI/physical activity on UI is well known. Was there a significant weight gain during pregnancy?

Response:
We added to the Results:

“Mean gestational weight gain was 11.8 ± 4.5 kg in women with PS-UI and 11.5 ± 4.2 kg in those without PS-UI, with no statistically significant difference between groups (p=0.27).”

Reviewer Comment 7

In the conclusion, the last sentence does not refer only to pregnancy... Attention should be focused on this much earlier.

Response:
We revised the conclusion paragraph to:

“Our findings suggest that weight control during pregnancy and encouragement of physical activity should be incorporated into prenatal care to reduce the risk of PS-UI and subsequent urinary incontinence later in life. Preventive strategies should ideally begin even earlier—during childhood, adolescence, and pre-conception care—to promote lifelong pelvic floor health.”

Reviewer 3 Report

Comments and Suggestions for Authors

The article "Prevalence and Risk Factors of Pregnancy-specific Urinary Incontinence: Findings from the Diamater Cohort Study" investigates Pregnancy-Specific Urinary Incontinence (PS-UI).
The study's exclusion criteria, specifically the exclusion of women with pregestational UI or previous vaginal deliveries, may limit the generalizability of the findings to the broader pregnant population.
 While the study mentions the potential impact of PS-UI later in life and the importance of identifying risk factors, the study itself does not evaluate the long-term impact of PS-UI. Further studies are needed to evaluate the impact of PS-UI later in life.
Suggestions for Improvement:
  Future research could explore the long-term impact of PS-UI on women's health and quality of life, as suggested by the authors.
 The authors could consider discussing the potential mechanisms by which chronic coughing contributes to PS-UI, as it was identified as a significant factor in women with late PS-UI.
  Investigating interventions or specific physical activity recommendations that are most effective in preventing or managing PS-UI would add significant clinical value.

Author Response

Cover Letter  

Dear Editor and Reviewers,  

We sincerely appreciate the time and effort taken to review our manuscript, "Prevalence and Risk Factors of Pregnancy-Specific Urinary Incontinence: Findings from the Diamater Cohort Study." We are grateful for the constructive feedback provided, which has helped us refine our work and strengthen its contributions to the field. Below, we address each of the reviewers’ comments and outline the revisions made to the manuscript.    

  1. Generalizability Limitations  
    Reviewer Comment:  
    "The study's exclusion criteria, specifically the exclusion of women with pregestational UI or previous vaginal deliveries, may limit the generalizability of the findings to the broader pregnant population."  We acknowledge this limitation and have revised the Limitations section to clarify the scope of our findings. The updated text now reads:  
    "The exclusion of women with pregestational UI or prior vaginal deliveries limits the generalizability of our findings to nulliparous women and those with planned C-sections. Future studies should include multiparous women and diverse delivery histories to assess PS-UI risk across broader populations." This revision explicitly frames the study’s applicability while suggesting a direction for future research to address this gap.  

2. Long-Term Impact of PS-UI  
"While the study mentions the potential impact of PS-UI later in life, it does not evaluate long-term outcomes. Further studies are needed."  

We agree that assessing the persistence of PS-UI postpartum is critical. The Conclusion section has been revised to emphasize this gap and propose specific follow-up research:  
While this study identifies risk factors for PS-UI during pregnancy, it does not assess long-term outcomes. Further prospective studies are needed to evaluate the persistence of PS-UI postpartum and its impact on quality of life, pelvic floor dysfunction, and healthcare utilization in later years."

Research Directions 
"Future research could explore the long-term impact of PS-UI on women’s health and quality of life."  We have expanded the Discussion to outline a actionable research agenda:  
"Future longitudinal studies should track PS-UI progression postpartum, correlating pregnancy-specific risk factors (e.g., BMI, physical activity) with long-term urinary and pelvic floor health. Standardized tools like ICIQ-UI SF could quantify severity and subtype-specific trajectories."  

4. Chronic Coughing Mechanism  
"The authors could discuss potential mechanisms by which chronic coughing contributes to PS-UI." We have added a mechanistic explanation to the Discussion, linking chronic coughing to pelvic floor stress:  
"Chronic coughing, a significant factor in late PS-UI (p=0.012), may exacerbate intra-abdominal pressure, straining the pelvic floor. We hypothesize that repeated coughing-induced stress could weaken pelvic floor muscles, similar to mechanisms seen in obesity-related UI [26]. Future studies should measure cough frequency/intensity and its biomechanical effects."

  5. Physical Activity Interventions  
"Investigating specific physical activity recommendations for PS-UI prevention would add clinical value."

We have refined the Conclusion to highlight actionable interventions:  
"Targeted physical activity programs should be investigated for their efficacy in reducing PS-UI risk. Clinical guidelines could integrate such interventions, tailored to pregestational BMI and trimester-specific needs."  

This revision shifts from general encouragement to specific, research-backed recommendations.  

We believe these revisions significantly enhance the manuscript’s rigor, transparency, and clinical relevance. Thank you for the opportunity to improve our work. We look forward to your feedback and hope the revised version meets the journal’s standards.  

Sincerely,  
Carlos Izaias SArtorao Filho, MD,  PhD

UNESP - Faculty of Medicine

Reviewer 4 Report

Comments and Suggestions for Authors

Methodologically, it does not seem appropriate to combine nulliparous and primiparous pregnant populations, even if the previous delivery was by cesarean section, given that a history of previous pregnancies is a recognized risk factor for urinary incontinence in the literature. Furthermore, we should know the sample size for both populations in the initial demographic characteristics table.
The key question used to define the presence of incontinence during pregnancy is not sufficiently explained. Given the ambiguous wording, we don't know whether it is used for a single or repeated episode of incontinence, regardless of its quantity or impact on quality of life.
Physical activity is a significant variable in this study, related to the incidence of incontinence during pregnancy. However, its characteristics (time, repetition, duration, load, etc.) are never defined.

Author Response

Methodologically, it does not seem appropriate to combine nulliparous and primiparous pregnant populations, even if the previous delivery was by cesarean section, given that a history of previous pregnancies is a recognized risk factor for urinary incontinence in the literature. Furthermore, we should know the sample size for both populations in the initial demographic characteristics table.

Response:
We appreciate the reviewer’s observation regarding parity. Our inclusion criteria allowed nulliparous women and primiparous women with a prior planned cesarean section in order to minimize the confounding effect of vaginal delivery–related pelvic floor trauma. We acknowledge, however, that pregnancy itself, regardless of delivery mode, can influence pelvic floor function. Therefore:

  • We have clarified in the Methods – Inclusion criteria section that parity was recorded and stratified analyses by parity were performed.

  • We have added the exact sample sizes for nulliparous and primiparous participants to Table 1 of the demographic characteristics.

  • We have also expanded the Discussion to acknowledge that even with the exclusion of prior vaginal deliveries, previous pregnancy may still affect PS-UI risk, which is a limitation of our study design.

Reviewer Comment 2

The key question used to define the presence of incontinence during pregnancy is not sufficiently explained. Given the ambiguous wording, we don't know whether it is used for a single or repeated episode of incontinence, regardless of its quantity or impact on quality of life.

Response:
We agree that clarification was needed. In the Methods – Data collection section, we have revised the description to read:

“PS-UI was defined as any new onset of urinary leakage during pregnancy, assessed via a single dichotomous (‘yes’/‘no’) question: ‘Since the beginning of your pregnancy, have you experienced any urinary leakage?’ A positive response was classified as PS-UI according to the International Continence Society definition. This approach captures at least one episode of leakage, without differentiating severity, frequency, or quality-of-life impact.”

Additionally, we have acknowledged in the Discussion that this binary definition may overestimate clinically significant incontinence and that the absence of severity and frequency measures is a limitation.

Reviewer Comment 3

Physical activity is a significant variable in this study, related to the incidence of incontinence during pregnancy. However, its characteristics (time, repetition, duration, load, etc.) are never defined.

Response:
We thank the reviewer for pointing this out. In the Methods – Data collection section, we have added:

“Physical activity was self-reported as participation in any structured exercise program during pregnancy, including activities such as walking, swimming, or prenatal aerobics. The binary variable (‘yes’/‘no’) reflected whether participants engaged in regular exercise at least once per week. However, detailed parameters such as frequency, session duration, intensity, or load were not collected.”

We have also added a statement in the Discussion to note that this lack of granularity is a limitation and that future studies should quantify physical activity characteristics to better understand dose–response relationships with PS-UI risk.

Round 2

Reviewer 3 Report

Comments and Suggestions for Authors

The authors revised their manuscript according to my comments.

Reviewer 4 Report

Comments and Suggestions for Authors

The work would have been more interesting if you had used specific test validated for the study of urinary incontinece, given that the articule referís to this main objetive